# The Allogenic Dental Pulp Transplantation from Son/Daughter to Mother/Father: A Follow-Up of Three Clinical Cases

**DOI:** 10.3390/bioengineering9110699

**Published:** 2022-11-17

**Authors:** Victor Pinheiro Feitosa, Mara Natiere Mota, Roseane Savoldi, Tainah Rifane, Diego de Paula, Livia Borges, Luzia Kelly Solheiro, Manoel Aguiar Neto, Lorena Vieira, Ana Carolina Moreira, Salvatore Sauro

**Affiliations:** 1Research Division, Paulo Picanço School of Dentistry, 900 Joaquim Sá St., Fortaleza 60135-218, Ceará, Brazil; 2Dental Biomaterials and Minimally Invasive Dentistry, Department of Dentistry, Cardenal Herrera-CEU University, CEU Universities, C/Santiago Ramón y Cajal, s/n., Alfara del Patriarca, 46115 Valencia, Spain

**Keywords:** pulp, regeneration, revitalization, transplant

## Abstract

The study investigated allogenic pulp transplantation as an innovative method of regenerative endodontic therapy. Three patients were selected for the endodontic treatment of single-root teeth, who also had a son/daughter with deciduous teeth or third molars scheduled for extraction. Receptor teeth were endodontically instrumented and irrigated using a tri-antibiotic solution. During the transplant procedures, the teeth from the son/daughter were extracted, sectioned, and the pulp was carefully removed. The harvested pulp from the donor was inserted into the root canal of the host tooth (father/mother), followed by direct pulp capping and resin composite restoration. The teeth were followed-up with for 2 years and were surveyed with computed tomography, the electric pulp vitality test, and Doppler ultrasound examination. At the 6-month follow-up, positive pulp vitality and the formation of periapical lesions were verified in cases 1 and 2. Case 3 showed remarkable periapical radiolucency before transplantation, but after 1 year, such lesions disappeared and there was positive vitality. All teeth were revascularized as determined by Doppler imaging after 2 years with no signs of endodontic/periodontal radiolucency. In conclusion, although this was a case series with only three patients and four teeth treated, it is possible to suppose that this allogenic pulp transplantation protocol could represent a potential strategy for pulp revitalization in specific endodontic cases.

## 1. Introduction

The pulp–dentin complex may be repaired, and dental caries can be arrested by highly organized, vascularized, and cell-loaded pulpal tissue. Any injuries to such tissue may trigger an inflammatory response or necrosis, overturning its repair capacity. Such circumstances often require pulp removal, disinfection, instrumentation, and endodontic obturation with inert materials such as gutta-percha [1]. Regenerative endodontics are rapidly evolving thanks to the advent of innovative tissue-engineering protocols, treatments, and technologies [2]. Most of the strategies used to regenerate pulp tissue are based on the use of stem cells from periodontal ligament, bone, or dental pulp. These cells are usually seeded in scaffolds and later transferred into the in-root canal, where they can differentiate in specialized cells towards the formation of vessels, fibroblasts, odontoblasts, and further cells found in pulp [3].

The role of growth factors is also essential for cell differentiation and signaling to the accessibility of other cells. Nevertheless, most of the improvements in regenerative endodontics are found in in vitro or animal model studies [2,3,4,5]. In this regard, the uses of platelet-rich fibrin (PRF) and platelet-rich plasma (PRP) have demonstrated optimal potential to treat open apex situations, improving the revascularization procedure [6,7,8,9]. However, the actual formation of new pulp tissue is limited when using platelet concentrates. Only recently, the transplantation of all the pulp tissue [10] generated an optimal “scaffold” for regeneration. The auto-transplantation of dental pulp collected from a third molar requiring extraction to a host tooth in the same patient was demonstrated to be a clinically feasible procedure for true pulp regeneration without risks for body rejection and severe inflammation [11,12,13]. Nevertheless, to the best of our knowledge, dental pulp transplantation between two different people has never been investigated so far. Indeed, allogenic transplantation could be the next step to assess the clinical success of such a clinical procedure.

Therefore, this investigation aimed at describing the clinical procedures for dental pulp allogenic transplantation and conducted a clinical follow-up for at least 24 months for three patients treated with such a regenerative endodontic treatment.

## 2. Materials and Methods

Three patients requiring root canal treatment in single-root premolars with their sons/daughters in need of an extraction of deciduous teeth or the third molar were recruited in this study (Dental Clinics of School of Dentistry). The protocol and the treatment plan were evaluated and approved by the Local Ethics Committee of Paulo Picanço School of Dentistry (protocol 3585776), and all patients signed an informed consent form with the details of the aim of the study and potential complications and accepted the clinical/radiographic follow-up for a period of 1, 3, 6, 12, 18, and 24 months. The inclusion criteria were the following: patients were 18–60 years old with no gender predilection, undergoing irreversible pulpitis (case 2) or pulp necrosis (cases 1 and 3) signs with spontaneous pain (case 2) or periapical radiolucency (cases 1 and 3) in single-root teeth, thereby requiring root canal treatment, with a periodontal pocket depth < 3 mm. The presence of tooth discoloration was not an exclusion criterion. Furthermore, the patients’ sons/daughters (donors) were aged between 9 and 21 years old and needed to have a deciduous tooth without root resorption or a third molar scheduled for extraction for reasons not related to this investigation. These donator teeth were tested for vitality with an electric pulp vitality test (Pulp Tester Digital, Odous de Deus, Belo Horizonte, Brazil).

First, panoramic radiography was performed on all patients in the three cases (Figure 1, Figure 2 and Figure 3), as well as computed tomography (CT) scanning with a 0.25 mm voxel size in order to confirm the periapical status of each tooth requiring endodontic treatment. The extraction of teeth from the patients’ children was performed without tooth sectioning and with extreme caution in order to avoid damaging the tooth and the apical structures. The extraction was carried out with local anesthesia using 1.8 mL of 2% lidocaine (1:80,000 epinephrine) through a nerve block technique. The extracted teeth were stored in a sterilized saline solution. At the same time and in the same dental office, the single-root teeth requiring root canal treatment were also anesthetized as aforementioned, isolated with a rubber dam, and pulp chamber access was executed with diamond burs with a high-speed handpiece under continuous and copious water irrigation. The canal instrumentation was performed using rotary files (Wave One Gold, Dentsply, Rio de Janeiro, Brazil) in combination with irrigation using a tri-antibiotic solution (ciprofloxacin, minocycline, and metronidazole at 500 μg/mL each) [14,15]. Prior to the rotary files, one #10 manual K-file (Dentsply) was used to perform the patency. No apical bleeding was performed because it was a different protocol than the revascularization strategy [15,16].

Sterile saline solution was used to remove the residues of the irrigating solution, while EDTA at 17% concentration was employed for 5 min for dentin conditioning and smear layer removal [17]. EDTA was rinsed twice using the saline solution, and the root canal was gently dried with an absorbent paper cone. The extracted teeth from the patients’ children were first cut with a small diamond saw with a slow-speed handpiece [10] under a running sterilized saline solution. This first cut penetrated only 1–2 mm around the entire mesial and distal surfaces and 3–4 mm along the occlusal surface. This was performed to create a notch without touching the pulp tissue. The notch was pressed with a straight sharp syndesmotome to separate the tooth into two halves, which allowed for more careful pulp removal using small tweezers [10]. Pulp tissue (from the donator tooth) was inserted into the root canal of the receptor tooth (mother or father) with the aid of disinfected gutta-percha cones [10] to avoid damaging the pulp tissue.

Direct pulp capping was performed by means of Biodentine (Septodont, Paris, France) calcium silicate cement, which was manipulated according to the manufacturer’s instructions and applied 2 mm below the bottom of the pulp chamber. After 10 min from the setting of the Biodentine, resin-modified glass ionomer cement (Riva Light-Cure, SDI, Bayswater, Australia) was applied as a liner and light-cured for 20 s with an LED Valo unit (1200 mW/cm^2^, Ultradent, South Jordan, UT, USA) in order to avoid a negative reaction between the Biodentine and adhesive restoration [18]. A two-step self-etch adhesive (Clearfil SE Bond, Kuraray Medical, Kurashiki, Japan) was applied in the selective enamel etching mode by using 37% phosphoric acid gel only at enamel borders for 15 s, which was rinsed with distilled water for 30 s, and both the dentin and enamel were completely air-dried for 30 s prior to the application of the adhesive primer for 20 s. A slight air-blast was performed to evaporate the solvent, followed by an immediate application of the adhesive for 20 s. The adhesive system was light-cured for 40 s using a Valo LED unit, as previously described. Resin composite build-up was incrementally performed using 2 mm thick increments (Opallis, FGM, Joinville, Brazil), which were individually light-cured for 20 s. After occlusal check/adjustments, the patient was sent home with pharmacological support based on 600 mg of ibuprofen every 8 h for 3 days. No antibiotics were prescribed for any patient. 

The receptor teeth were imaged by computed tomography in this first clinical visit (Figure 4(A1,B1,C1,C2)). Patients were scheduled for follow-up visits after 1 month and at every 3 months after the allogenic transplantation clinical procedure. At each recall visit, the treated teeth were evaluated via computed tomography (at 6, 12, and 24 months) and periapical radiography (at 3, 9, 15, and 18 months). Moreover, an electric pulp vitality test (Pulp Tester Digital, Odous de Deus, Belo Horizonte, Brazil) was performed on treated and further intact teeth from the same patient for control purposes, followed by occlusal check/adjustments of restorations and Doppler ultrasonic imaging at the 2-year follow-up. The device used was a Phillips HD11 ultrasonic device with color and spectral Doppler with a 1 cm/s rate by using a linear tip with 12–15 MHz frequency.

## 3. Results

During the 3-month follow-up, all patients reported slight pain at the periapical region of the receptor tooth, which started after 40 days and continued up to 2–3 months after the procedure. No response to the electrical pulp vitality test was detected until the 3-month follow-up. The patients in cases 1 and 2 showed a slight increase in periapical radiolucency as highlighted in the computed tomography at the 6-month follow-up period (Figure 4(A2,B2)) and at 1 year in case 1 (Figure 4(A3)). The details of patients from each case are presented in Table 1.

After one year from the allogenic transplantation, a complete regression of periapical lesions was verified for the patient in case 2 (Figure 4(B3)), and a remarkable reduction was verified for the patient in case 3 (Figure 4(C3,C4)). No signs of endodontic/periodontal complications/radiolucencies were observed for any patients at the 24-month follow-up stage (Figure 4(A4,B4,C5,C6)). Positive pulp vitality was confirmed, and revascularization was further proved for the three patients by Doppler imaging (Figure 4(D1–D4)) at the periapical region of each tooth.

## 4. Discussion

In modern dentistry, the auto-transplantation of teeth is a well-known and established procedure for the replacement of destroyed molars demanding immediate extraction. More recently, 3D-printing technology facilitated such a procedure by allowing the training of operators in a simulated clinical scenario [19]. Even the allogenic tooth transplantation of an entire tooth from the daughter to the mother was successfully demonstrated with the support of 3D-printing technology in order to achieve an optimal clinical outcome [20]. Such clinical procedures prompted the concept and idea of the present investigation, which was based on the use of the pulp tissue from an extracted primary tooth to replace the defective tissue in a different area of the oral environment in a third patient.

Several advantages are expected when performing a pulp tissue allogenic transplantation, such as the reduction of transplant rejection, similar DNA and RNA in all cells, completely mature connective tissue, formed neuronal network, as well as a properly formed vascularization network. Nevertheless, small inflammation would be expected at the initial time until the macrophages and lymphocytes recognize similar DNA fragments diminishing immunological reactions, thereby avoiding complete tissue rejection and the unsuccess of transplants [20]. Indeed, these factors may enhance the success of pulp regeneration treatments as in the strategy demonstrated by Liang et al. [13], which used minced pulp as a source of mesenchymal stem cells (MSCs) in order to provide odontogenic differentiation with high potential for protein and alkaline phosphatase expression. Furthermore, the in vivo clinical regeneration of pulp tissue would be faster than when using only conventional scaffolds, MSCs, and growth factors, especially once all the conjunctive tissue, blood vessels, and nerves are already constructed.

In a successful clinical procedure for pulp allogenic transplantation, the extraction of primary teeth or third molars from a relative, such as sons or daughters, must be performed with minimal damage of the tooth, avoiding tooth sectioning during extraction and excluding carious teeth or resorbed roots.

Indeed, when pulp tissue was removed from the donator teeth with tweezers, several odontoblasts were heavily damaged but were suitably replaced by stem cells after transplantation in order to achieve clinical success. Nevertheless, such a procedure is critical, as many cells could be damaged or killed, depending on how careful the pulp removal from the pulp chamber in the donor teeth was. A harsh procedure may inevitably cut the cellular processes from the odontoblast cell body, thereby increasing the failure rate of the entire transplantation procedure. In a therapeutic role, alkaline phosphatase may act as recovery mechanisms to heal the damaged tissue with odontoblast-like cells or with localized pulp mineralization [13].

Herein, the primary teeth extracted (asterisks in Figure 1 and Figure 2) were scheduled to be extracted with minimal injury as they were vertically inclined, and permanent teeth were already formed/erupted in cases 1 and 2. Furthermore, the tooth must be stored as soon as possible in a sterilized saline solution; the complete disinfection prior to pulp removal and transplantation is mandatory [21]. The sectioning of the extracted tooth must be performed using a sterilized handpiece and diamond disks only after the root canal instrumentation of the receptor tooth [10]. Pulp should also be removed with sterilized tweezers (Figure 4B) and inserted into the roots of the acceptor teeth using disinfected gutta-percha cones. Indeed, all these procedures may promote minimal contamination during the clinical treatment, thereby increasing the probability to perform bacterial-free endodontic regeneration.

Moreover, root canal instrumentation and irrigation were performed by using rotary files and a tri-antibiotic solution [22], respectively. Although a recent report [23] suggested that clindamycin can replace minocycline to offer better angiogenic potential, we in the present case series employed the traditional solution used for revascularization procedures. A further feasible drug to replace minocycline would be doxycycline [24], which is approved by the US FDA and does not induce oxidative tooth staining over time [25]. However, doxycycline has been little investigated in pulp regeneration studies and the tri-antibiotic solution was used only as an irrigant, thereby diminishing the possibility of tooth staining. This is in contrast with the conventional procedure of revascularization performed by endodontic specialists, which requires keeping the tri-antibiotic solution for several days inside the root canals and pulp chamber. A further traditional procedure commonly performed in such a scenario is a final irrigation with 17% EDTA [17] prior to tissue transplantation [10]. It has been proposed that by opening dentinal tubules and conditioning the outer dentin, it would be possible to release several growth factors such as brain-derived neurotrophic factors and growth-differentiation factor 15 [8]. Particularly, these neuronal growth factors play an important role in the regeneration process of the transplanted pulp and obtain a faster recovery of the tooth vitality [26]. Furthermore, the intense blood flow in the bone surrounding the periapical apex may deliver vascular endothelial growth factors (VEGF) into root canals, likely reaching the two receptors (VEGFR-1 and VEGFR-2) in cells from mature blood vessels [26] within the transplanted pulp tissue. With entire vessels formed inside and outside the root canal in the apical region, a small vascular formation would restore blood to the transplanted pulp. All apexes from receptor teeth were kept without a high increase in diameter, but an open apex could help in revascularization and clinical success.

Concerning the few clinical studies available in the literature about pulp regeneration, the majority of them are focused on teeth diagnosed with irreversible pulpitis [8,9,12] with an absence of initial periapical radiolucency; this provides optimal vascularization and a lack of inflammatory mediators in the bone of the periapical region, which may facilitate the revascularization and reinnervation of transplanted tissue/cells. Conversely, in the current clinical cases, all patients presented periapical radiolucency either before or a few months after the allogenic transplantation. Such a feature may be a consequence of lower blood irrigation along with some inflammatory mediators due to the presence of allogenic cells, which may interfere with the success of the treatment. Nevertheless, successful outcomes were attained in the current study, along with a clear reduction of the periapical lesions in the treated teeth (Figure 4). Herein, we did not use prefabricated fibrin or collagen to avoid the contact of stem cells with different tissues, which could impair their differentiation and the success rate of transplantation. Stem cells replicate and differentiate more when in their natural tissue; this is the reason we did not use such materials.

Several protocols have been proposed for pulp transplantation to the root canal via endodontic regeneration, such as minced pulp [13], extracted/expanded DPSCs [12], platelet-rich fibrin [8], and leukocyte-platelet-rich fibrin [9]. Unfortunately, all these treatments require the support from a laboratory procedure to obtain platelet concentrates or stem cell expansion/replication [21]. In the present investigation, no laboratory intervention was demanded for pulp allogenic transplantation, which may represent a significant advance in comparison to previous clinical strategies. In fact, pulp allogenic transplantation as highlighted in the present report is entirely performed in a clinical dental office without the need for further external support.

In the clinical steps of the present case series, pulp capping after the allogenic transplantation was performed by using a gold standard procedure based on the use of calcium silicate cement (Biodentine, Septodont). This usually attains an optimal pulpal response, yielding a release of odontogenic growth factors, alkaline phosphatase release, and dentin bridge formation [9,27]. Afterwards, resin-modified glass ionomer cement was applied, followed by the application of [28] a two-step self-etch adhesive Clearfil SE Bond, which was employed in the selective enamel etching technique [29] in order to obtain the optimal sealing of the margins of the restored cavity. We believe that such pulp capping and restorative procedures may play an important role in the clinical long-term outcome of auto-transplanted pulps.

To confirm the apical revascularization, the three patients received a Doppler ultrasonic evaluation (Figure 4) of the four teeth that were treated for a pulp allogenic transplantation. This technique confirmed blood perfusion with characteristic pulse images (Figure 4(D4)), thereby demonstrating revascularization. Although such a method is not usual for the assessment of pulp vitality and periapical vascularization, an experienced M.D. radiologist trained with an obstetric ultrasound/Doppler device using his own teeth until it was calibrated prior to surveying the patients. Furthermore, electric pulp testing may vary from one device to another concerning the level of electric current and the tips employed, thereby jeopardizing the repeatability and reliability of the experiment.

This report is the very first attempt to demonstrate that the transplantation of pulp may be possible between close relatives such as parents and children, and perhaps between further family members. Indeed, future trends should focus on possible blood tests to survey the suitability for pulp transplants among distant family members and non-family-related donors/receptors. However, there are several limitations to the present treatment protocol related to the treatment of cases characterized by teeth presenting abscesses, purulency, or even when third molars are not well-positioned, which need to be extracted employing odonto-section procedures with a high risk for pulp damage and/or contamination. Furthermore, more improvements for the clinical procedure should focus on more suitable irrigation protocols and antibiotics, optimal root canal instrumentation, and the digitalization of the process in order to facilitate the current positioning of the pulp tissue into root canals with different shapes and sizes. Future trends may continue with the investigation of the health status of transplanted tissue by histochemical staining or cell culture studies, as well as the possible assessment of pulp transplantation between people from different families and a future feasible test to confirm gene compatibility tailored to such a novel sort of transplantation procedure.

## 5. Conclusions

Within the limitations of the present clinical report, the procedure of allogenic pulp transplantation may be feasible and provide revascularization, vitality restoration, and the reduction of periapical radiolucency. Further randomized clinical trials should be employed to study the actual success rate of such an innovative clinical treatment.

## Figures and Tables

**Figure 1 bioengineering-09-00699-f001:**
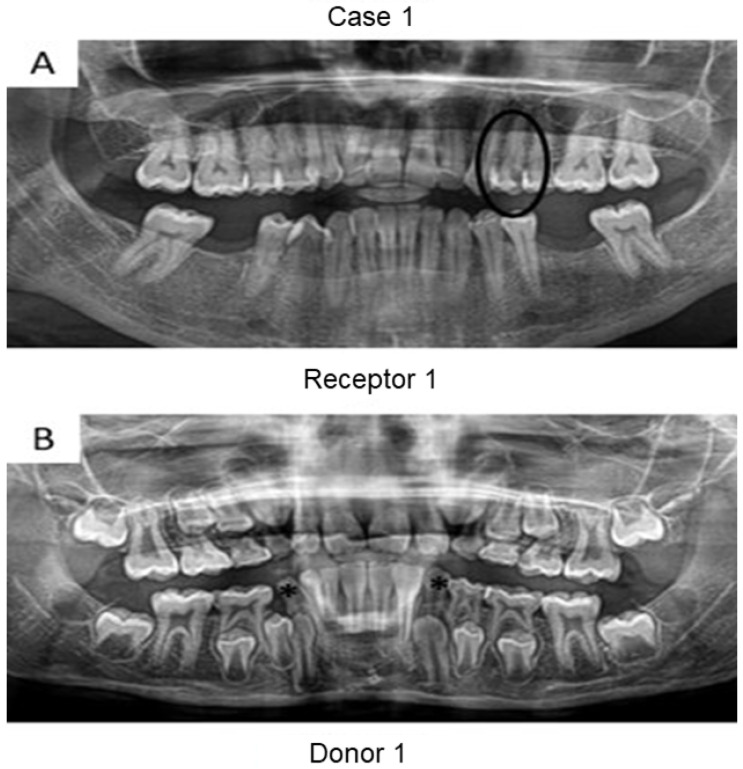
Panoramic radiographs of initial visit of the two patients from case 1. Black circle indicates the tooth requiring root canal treatment (receptor tooth) in the mother whilst the asterisks point to the primary canines in the son with planning for extraction (pulp donator tooth). The pulps from both canines were transplanted to the buccal and palatal canals from the first premolar of the mother.

**Figure 2 bioengineering-09-00699-f002:**
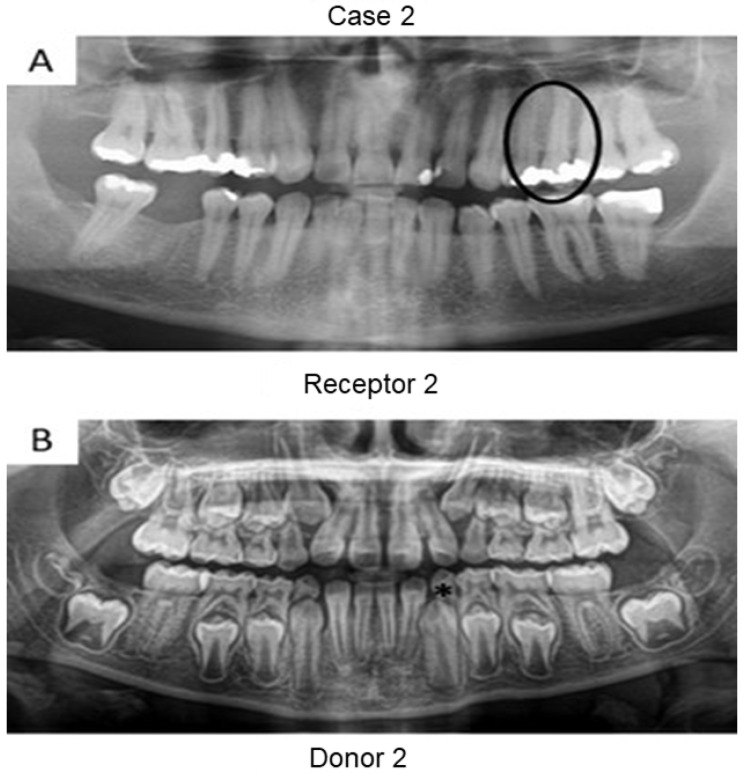
Panoramic radiographs of initial visit of the two patients in case 2. Black circle indicates the tooth requiring root canal treatment (receptor tooth) in the father whilst the asterisk points to the primary canine of the patient’s daughter planned for extraction (pulp donator tooth).

**Figure 3 bioengineering-09-00699-f003:**
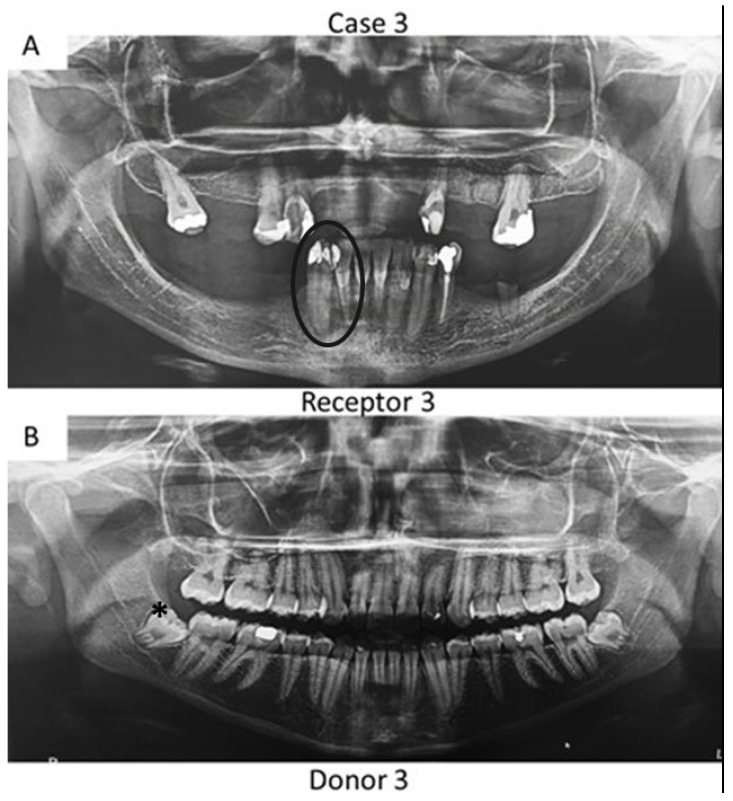
Panoramic radiographs of initial visit of the further two patients. Black circle indicates the teeth requiring root canal treatment (receptor teeth: lateral incisor and canine) in the mother whilst the asterisk highlights the third molar from the daughter demanding extraction (pulp donator tooth). The pulp from the donator tooth was sliced to obtain a mesial portion, which was transplanted to the lateral incisor of the mother; meanwhile, the distal part of the pulp was inserted into the mother’s canine.

**Figure 4 bioengineering-09-00699-f004:**
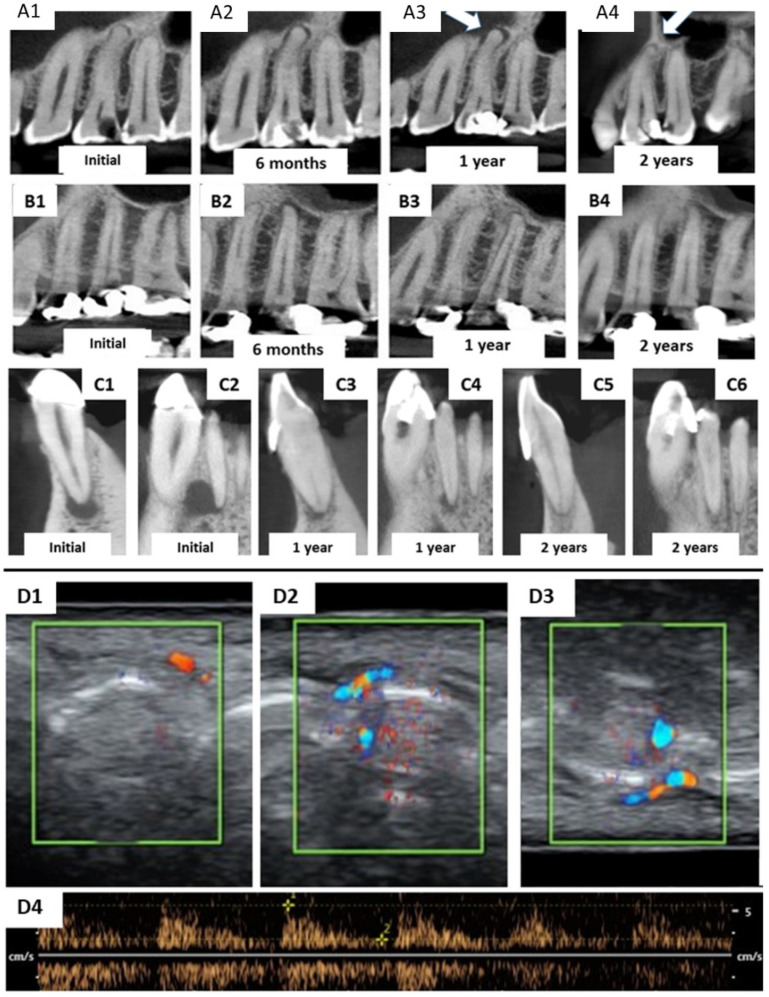
Computed tomography images of teeth receiving pulp allogenic transplantation. (**A1**–**A3**) are related to the patient’s premolar in case 1, which showed small initial periapical radiolucency (**A1**) and slightly increased in further follow-ups (**A2**,**A3**) but was entirely reduced at the 2-year follow-up (**A4**). The same trend occurred in the patient from case 2 (**B1**,**B2**) with a complete reduction of the periapical lesions at the 1-year follow-up (**B3**). This was confirmed by tomography at the 2-year stage (**B4**). In case 3, two teeth received allogenic pulp transplantation: the lateral incisor and mandibular canine. However, different tomographic slices were selected to show the treatment progression; (**C1**,**C3**,**C5**) depict lesion regression at the 1- and 2-year follow-ups. The same occurred with the lateral incisor, with a complete reduction of periapical radiolucency after 2 years. All four teeth from the 3 cases showed positive pulpal vitality with the electric test after 6–12 months. (**D1**–**D4**) depicts the Doppler ultrasonic assessment of the periapical revascularization in teeth from all three cases subjected to pulp allogenic transplantation. (**D1**) demonstrates high blood flux at the tooth receptor of the pulp of the patient from case 1. The blood flux of the teeth from case 2 (**D2**) was intense and similar to that of both teeth in case 3 (**D3**). The pulse intensity was normal in all four teeth (**D4**).

**Table 1 bioengineering-09-00699-t001:** Spreading of age and gender of donors and receptors in each case along with the teeth related.

	Age	Relationship	Teeth
Donor 1	11	Son	73
Receptor 1	25	Mother	24
Donor 2	10	Daughter	73
Receptor 2	49	Father	25
Donor 3	20	Daughter	38
Receptor 3	48	Mother	42 and 43

## Data Availability

No new data were created or analyzed in this study. Data sharing is not applicable to this article.

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
