# Peer review of "The Allogenic Dental Pulp Transplantation from Son/Daughter to Mother/Father: A Follow-Up of Three Clinical Cases"

_bioengineering, 2022, doi:10.3390/bioengineering9110699_

Round 1
Reviewer 1 Report (New Reviewer)
Current work on dental pulp transplantation by Feitosa et al. is a further step based on their previous report "Dental Pulp Autotransplantation: A New Modality of Endodontic Regenerative Therapy—Follow-Up of 3 Clinical Cases. JOE 2021". The idea is novel, and the results are encouraging. I would like to suggest accept of this paper.
Author Response
Reviewer 1
Current work on dental pulp transplantation by Feitosa et al. is a further step based on their previous report "Dental Pulp Autotransplantation: A New Modality of Endodontic Regenerative Therapy—Follow-Up of 3 Clinical Cases. JOE 2021". The idea is novel, and the results are encouraging. I would like to suggest accept of this paper.
Answer: We thank the referee for the comments.
Reviewer 2 Report (New Reviewer)
The case series report is interesting for clinicians and reserchers, but I am concerned about these treatments from ethical point of view.
The authors of this article reported dental pulp autotransplantation by same methods in 2021 (References No. 10). They have attempted the allogenic dental pulp transplantation in this article. The results are innovative, but there are several questions.
At first, the therapeutic procedure in the present case report is same as the previous case report (References No. 10), but the source of the pulp tissues are totally different form previous one. Nevertheless, the protocol and treatment plan were approved as the same protocol No. (3585776) in both reports. Furthermore, legal or formal nomenclature of the ethics committee is not described.
Next, Cone Beam Computed Tomography (CBCT) provides many information about dentition and skeleton, whereas radiation dose for CBCT is much higher than intraoral radiography. Hence, use of CBCT in endodontics is limited. American Association of Endodontists (AAE) and American Academy of Oral and Maxillofacial Radiology (AAOMR) present the statement for use of CBCT in endodontics, and it is standard for not only endodontists but also dentists. In the statement following, the following contents are specified; “CBCT should be used only when the patient’s history and a clinical examination demonstrate that the benefits to the patient outweigh the potential risks.” “CBCT should not be used routinely for endodontic diagnosis or for screening purposes in the absence of clinical signs and symptoms.” “Clinicians should use CBCT only when the need for imaging cannot be met by lower dose two-dimensional (2D) radiography.” The authors completely violated the statement.
Furthermore, the detail of the diagnosis is not described. They simply described as “requiring root canal treatment”. Especially, the necessity for pulpectomy in Case 2 is unclear.
Authors should describe more detail in revised article to respond to these concerns.
Author Response
Reviewer 2
The case series report is interesting for clinicians and reserchers, but I am concerned about these treatments from ethical point of view.
The authors of this article reported dental pulp autotransplantation by same methods in 2021 (References No. 10). They have attempted the allogenic dental pulp transplantation in this article. The results are innovative, but there are several questions.
At first, the therapeutic procedure in the present case report is same as the previous case report (References No. 10), but the source of the pulp tissues are totally different from previous one. Nevertheless, the protocol and treatment plan were approved as the same protocol No. (3585776) in both reports. Furthermore, legal or formal nomenclature of the ethics committee is not described.
Answer: The correct name of Ethics Committee was included in the manuscript. The project submitted and approved by the Committee encompassed both autologous and allogenic transplants.
Modified text: “The protocol and the treatment plan were evaluated and approved by the local Ethics Committee of Paulo Picanço School of Dentistry (Protocol 3585776)”
Next, Cone Beam Computed Tomography (CBCT) provides many information about dentition and skeleton, whereas radiation dose for CBCT is much higher than intraoral radiography. Hence, use of CBCT in endodontics is limited. American Association of Endodontists (AAE) and American Academy of Oral and Maxillofacial Radiology (AAOMR) present the statement for use of CBCT in endodontics, and it is standard for not only endodontists but also dentists. In the statement following, the following contents are specified; “CBCT should be used only when the patient’s history and a clinical examination demonstrate that the benefits to the patient outweigh the potential risks.” “CBCT should not be used routinely for endodontic diagnosis or for screening purposes in the absence of clinical signs and symptoms.” “Clinicians should use CBCT only when the need for imaging cannot be met by lower dose two-dimensional (2D) radiography.” The authors completely violated the statement.
Answer: The referee’s concerns are correct. However, the CBCT were performed almost once a year, exactly to avoid such over exposure to radiation. Intraoral radiographies were used in most recalls and the entire protocol for recalls was approved both by the Ethics Committee and by the patients.
Furthermore, the detail of the diagnosis is not described. They simply described as “requiring root canal treatment”. Especially, the necessity for pulpectomy in Case 2 is unclear.
Authors should describe more detail in revised article to respond to these concerns.
Answer: The detailed description of pulp diagnosis was included in the manuscript as required. In case 2, the patient came for the dental school with spontaneous pain and all tests detected irreversible pulpitis.
Modified text: “Inclusion criteria were patients among 18-60 years old with no gender predilection, undergoing irreversible pulpitis (case 2) or pulp necrosis (cases 1 and 3) signs with spontaneous pain (case 2) or periapical radiolucency (cases 1 and 3) in single-root teeth, thereby requiring root canal treatment”
Reviewer 3 Report (New Reviewer)
Dear editor,
Thank you for inviting me to review this manuscript. I congratulate the authors and think it is a very interesting report of three cases with successful allogenic transplantation of dental pulp. The manuscript is well-written and I can only suggest some minor modifications.
1- I suggest adding two parts to the discussion. First, what is your hypothesis on the underlying mechanism of this revascularization? Since it is not easy to imagine how the transplanted tissue received blood supply and regenerated root canal pulp. Second, please add some literature review on previous allogenic transplantations and how the immunologic reactions could be related to apical radiolucency.
2- Did you perform any blood test to relate the differences between parents and children to the immunologic responses? If not, please discuss and suggest if you think it is relevant.
3- There are some typos throughout the text, such as kids (instead of using children).
Author Response
Reviewer 3
Dear editor,
Thank you for inviting me to review this manuscript. I congratulate the authors and think it is a very interesting report of three cases with successful allogenic transplantation of dental pulp. The manuscript is well-written and I can only suggest some minor modifications.
1- I suggest adding two parts to the discussion. First, what is your hypothesis on the underlying mechanism of this revascularization? Since it is not easy to imagine how the transplanted tissue received blood supply and regenerated root canal pulp. Second, please add some literature review on previous allogenic transplantations and how the immunologic reactions could be related to apical radiolucency.
Answer: The insights were included in the Discussion section as required. Also, references 20 (a literature review) and 26 were included to support allogenic transplantation immunologic reactions and revascularization.
Modified text: “Even allogenic tooth transplantation of entire tooth from the daughter to the mother was successfully demonstrated with support of a 3D-printing technology in order to achieve optimal clinical outcome [20]. … similar DNA and RNA in all cells, completely mature connective tissue, formed neuronal network as well as a properly formed vascularization network. Nevertheless, small inflammation would be expected at initial times, until the macrophages and linfocytes recognizes similar DNA fragments diminishing immunological reactions, thereby avoiding complete tissue rejection and insuccess of transplants [20]. ” and
“Furthermore, the intense blood flow in bone surrounding periapical apex may induce the income of vascular endothelial growth factors (VEGF) into root canal, likely reaching the two receptors (VEGFR-1 and VEGFR-2) in cells from mature blood vessels [26] within the transplated pulp tissue. With the entire vessels formed within and outside root canal in apical region, small vascular formation would induce late blood regain to transplanted pulp.”
2- Did you perform any blood test to relate the differences between parents and children to the immunologic responses? If not, please discuss and suggest if you think it is relevant.
Answer: No blood tests were performed. This issue was included in Discussion section.
Modified text: “Indeed, future trends should focus on possible blood tests to survey suitability for pulp transplants among distant family members and not family-related donor/receptor ones.”
3- There are some typos throughout the text, such as kids (instead of using children).
Answer: All typos were corrected and “kids” changed to “children”
Reviewer 4 Report (New Reviewer)
Dear authors,
thank you for the opportunity to review the manuscript dealing with an interesting clinical idea.
However, the article and in particular the M&M, Results and discussion sections are lacking some crucial points, as listed below:
Abstract:
L. 23: "selected teeth" is confusing, as it could refer to the donor or acceptor teeth.
L. 29: formation of periapical lesion is unclear, as it may lead to the conlcusion that these 2 teeth (and therefore 66% of the study population) were a nonsuccess.
Introduction:
L. 40: "Pulp is connected to the human 40 body and with further organs through the apical foramen, " is generally known and should be rephrased or removed.
L.42: Please don't use and together with or.
L. 45: "Regenerative endodon-45 tics is " should be "are". Please double-check the grammar and language of the manuscript.
Materials and Methods:
Please list the inclusion criteria for donors and acceptors, seperately. Have you checked the vitality of the donor teeth? How old were the children and the parents? Right before exfoliation of the deciduous teeth, there might be some serious resorption in the area of the apical tissue so informations about the donor teeth would be crucial. During the whole Materials and methods part you skip confusingly between donor and acceptor. To the reader it is unclear, if both were in the dental office at the same time or even in the same room.
L. 85: "but with extreme caution in 85 order to avoid damaging the tooth and the apical structures and (in case 3). " seems to be mal-formatted. Please check.
L. 93: How long was the exposure time of the tri-antibiotic solution and in which concentration were the three antibiotics applied?
L. 95: Sentence "No apical bleeding was performed 95 because it was a different protocol from revascularization strategy (15,16) " is unclear and should be removed.
L. 128: Typo "recall visi". Please carefully check your manuscript.
Results:
Figures: Have you also performed some standrad dental films? These are the common standard for apical diagnosis as they provide a better resluton than CBCTs. You should add some standard dental films or give at least some information about the CBCT, especially the voxel size. Furthermore, some of the X-rays differ substantially regarding their axis and 3-dimensional orientation. For the Doppler analysis, basline images would be crucial.
DIscussion:
L. 223: Revascularization is not a conventional procedure, as it is not farly distributed and should be done by specialists.
Your discussion is lacking the folloing points:
-Does the apex need to be open for successful transplantation?
-Why didnt you use prefabricated fibrin or collagen materials in addition to the pulp?
-Odontoblasts and their processes reaching into dentin are surely heavily damaged, when they are removed from a pulp using tweezers. You should discuss the possible mechanisms during transplantation, as odontoblasts can normally not be removed or cultured in a vital state.
Author Response
Reviewer 4
Dear authors,
Thank you for the opportunity to review the manuscript dealing with an interesting clinical idea.
However, the article and in particular the M&M, Results and discussion sections are lacking some crucial points, as listed below:
Abstract:
- 23: "selected teeth" is confusing, as it could refer to the donor or acceptor teeth.
Answer: We changed to “receptor teeth”.
- 29: formation of periapical lesion is unclear, as it may lead to the conclusion that these 2 teeth (and therefore 66% of the study population) were a nonsuccess.
Answer: Indeed, the referee is correct, but two lines below it was clear that those lesions disapperead at 1-year recalls.
Introduction:
- 40: "Pulp is connected to the human 40 body and with further organs through the apical foramen, " is generally known and should be rephrased or removed.
Answer: The sentence was removed.
Modified text:”Pulp-dentin complexes may be repaired and dental caries can be arrested by the highly organized, vascularized, and cell-loaded pulpal tissue. Any injuries to such tissue may trigger an inflammatory response or necrosis, overturning its repair capacity. Such a circumstance it is often required pulp removal, disinfection, instrumentation, and endodontic obturation with inert materials such as gutta percha [1]”
L.42: Please don't use and together with or.
Answer: We corrected the term. Thanks!
- 45: "Regenerative endodontics is " should be "are". Please double-check the grammar and language of the manuscript.
Answer: The grammar and language of the manuscript were revised and modified.
Modified text: “Regenerative endodontics are rapidly evolving thanks to the advent of innovative tissue engineering protocols, treatments, and technologies [2]”
Materials and Methods:
Please list the inclusion criteria for donors and acceptors, seperately. Have you checked the vitality of the donor teeth? How old were the children and the parents? Right before exfoliation of the deciduous teeth, there might be some serious resorption in the area of the apical tissue so informations about the donor teeth would be crucial. During the whole Materials and methods part you skip confusingly between donor and acceptor. To the reader it is unclear, if both were in the dental office at the same time or even in the same room.
Answer: The issues were included in the manuscript.
Modified text: “Furthermore, the patients’ sons/daughters (donors) were aged between 9 and 21 years-old and needed to have a deciduous tooth without root resorption or a third molar scheduled for extraction for reasons not related to this investigation. These donator teeth were tested to vitality with electric pulp vitality test (Pulp Tester Digital, Odous de Deus, Belo Horizonte, Brazil).”
and
“The extraction was carried out in local anesthesia using 1.8 mL of 2% lidocaine (1:80,000 epinephrine) through nerve block technique. The extracted teeth were stored in sterilized saline solution. In the same time and in the same dental office, the single-root teeth requiring root canal treatment were also anesthetized as aforementioned, isolated with rubber dam and pulp chamber access was executed with diamond burs in high-speed handpiece under continuous and copious water irrigation. The canal instrumentation was performed using rotary files (Wave One Gold”
- 85: "but with extreme caution in 85 order to avoid damaging the tooth and the apical structures and (in case 3). " seems to be mal-formatted. Please check.
Answer: The sentence was revised.
Modified text: “The extraction of teeth from patients’ children was performed without tooth-sectioning, and with extreme caution in order to avoid damaging the tooth and the apical structures.“
- 93: How long was the exposure time of the tri antibiotic solution and in which concentration were the three antibiotics applied?
Answer: The tri-antibiotic paste was applied as an irrigant during endodontic treatment, so the time was dependent to the duration of instrumentation procedure. The concentration of antibiotics was 500 micrograms per mL for each antibiotic, similar to that of the reference quoted.
- 95: Sentence "No apical bleeding was performed 95 because it was a different protocol from revascularization strategy (15,16) " is unclear and should be removed.
Answer: In the revascularization protocol, decontamination of the canal is performed, irrigation, insertion of an intracanal medication, and the induction of bleeding that will form a clot. In the study, this method was not applied, showing the importance of containing this information.
- 128: Typo "recall visi". Please carefully check your manuscript.
Answer: All typos and grammar mistakes was revised.
Results:
Figures: Have you also performed some standrad dental films? These are the common standard for apical diagnosis as they provide a better resluton than CBCTs. You should add some standard dental films or give at least some information about the CBCT, especially the voxel size. Furthermore, some of the X-rays differ substantially regarding their axis and 3-dimensional orientation. For the Doppler analysis, basline images would be crucial.
Answer: The standard x-rays were performed in some recalls, but CBCTs provided more detailed images in the present cases. The details were included in the text. However, baseline (initial) Doppler images are missing and could not be provided at this point.
Discussion:
- 223: Revascularization is not a conventional procedure, as it is not farly distributed and should be done by specialists.
Answer: The referee is correct. We corrected the sentece to make clear it is usual to endodontic specialists.
Modified text: “This is in contrast with the conventional procedure of revascularization performed by endodontic specialists, which requires to keep the triantibiotic solution for several days inside the root canals”
Your discussion is lacking the following points:
-Does the apex need to be open for successful transplantation?
Answer: The issue was included in Discussion section.
Modified text: “Furthermore, the intense blood flow in bone surrounding periapical apex may induce the income of vascular endothelial growth factors (VEGF) into root canal, likely reaching the two receptors (VEGFR-1 and VEGFR-2) in cells from mature blood vessels [26] within the transplated pulp tissue. With the entire vessels formed within and outside root canal in apical region, small vascular formation would induce late blood regain to transplanted pulp. All apex from receptor teeth were kept without high increase in diameter, but an open apex could help in revascularization and clinical success.
-Why didnt you use prefabricated fibrin or collagen materials in addition to the pulp?
Answer: We included this issue in Discussion.
Modified text: “We did not use prefabricated fibrin or collagen to avoid contact of stem cells with different tissues, that could impair their differentiation and success rate of transplantation. Stem cells replicate and differentiate more when in their natural tissue, this is the reason we did not use such materials.”
-Odontoblasts and their processes reaching into dentin are surely heavily damaged, when they are removed from a pulp using tweezers. You should discuss the possible mechanisms during transplantation, as odontoblasts can normally not be removed or cultured in a vital state.
Answer: The issue was included in Discussion as required.
Modified text: “In a successful clinical procedure for pulp allogenic transplantation, the extraction of primary teeth or third molars from a relative such as sons or daughters must be performed with minimal damage of the tooth, avoiding tooth sectioning during the extraction and excluding carious teeth or with resorbed roots. Indeed, when pulp tissue was removed from donator teeth with tweezers, some odontoblasts were heavily damaged, but with suitable replacement by stem cells after transplants in order to achieve clinical success.”
We hope these changes turn the paper acceptable for publication.
Best regards,
Authors
Round 2
Reviewer 2 Report (New Reviewer)
The authors answered for questions and revised the descriptions.
I can agree to accept this article.
Author Response
We thank the referee for accepting the manuscript.
Reviewer 4 Report (New Reviewer)
Dear authors,
thank you for the renewed opportunity to review the interesting manuscript.
It has been improved substantially since the last review round.
However, there are still some points that need adaptation:
-Please add a table showing the patient characteristics (age, gender etc) of the donor patients and teeth as well as the receptor patients and teeth.
-Add the information about the antibiotic conenctration in the rebuttal letter to the manuscript.
-I disagree with the point that "some odontoblasts were heavily damaged". Be removing the entire pulp you inevitably cut the cellular processes from the odontoblast cell body, so all odontoblasts are at least damaged or more probable even killed. Therefore, you should thoroughly add a paragraph discussing the potential mechanisms of your procedure in the introduction and the discussion part.
Author Response
The authors thank the Editor and referees of the Bioengineering MDPI for their constructive comments. The changes in the text are highlighted in yellow and by using “track changes”. Below you can find the answers to each comment of referees.
Reviewer 4
Dear authors,
thank you for the renewed opportunity to review the interesting manuscript.
It has been improved substantially since the last review round.
However, there are still some points that need adaptation:
-Please add a table showing the patient characteristics (age, gender etc) of the donor patients and teeth as well as the receptor patients and teeth.
Answer: The Table was included in Results section.
-Add the information about the antibiotic conenctration in the rebuttal letter to the manuscript.
Answer: The concentration was included in the manuscript.
-I disagree with the point that "some odontoblasts were heavily damaged". Be removing the entire pulp you inevitably cut the cellular processes from the odontoblast cell body, so all odontoblasts are at least damaged or more probable even killed. Therefore, you should thoroughly add a paragraph discussing the potential mechanisms of your procedure in the introduction and the discussion part.
Answer: The paragraph was added as follows.
Modified text: “Indeed, when pulp tissue was removed from donator teeth with tweezers, several odontoblasts were heavily damaged, but with suitable replacement by stem cells after transplants in order to achieve clinical success. Nevertheless, such procedure is critical, as many cells could be damaged or killed, depending on how carefull was the pulp removal from pulp chamber in donor teeth. A harsh procedure may inevitably cut the cellular processes from the odontoblast cell body, thereby achieving likely insuccess to the entire transplantation procedure.“
We hope these changes turn the paper acceptable for publication.
Best regards,
Authors
This manuscript is a resubmission of an earlier submission. The following is a list of the peer review reports and author responses from that submission.
Round 1
Reviewer 1 Report
Feitosa V et al.: The allogenic dental pulp transplantation from son/daughter to 2 mother/father: a follow-up of three clinical cases
This article provides three clinical cases of allogenic dental pulp transplantation. The provided successful cases may be informative for developing the innovative method of regenerative endodontic therapy. However, there are major concerns in the following items. Thus, this study should be improved before consideration for publication in Bioengineering.
Major concerns:
- The authors describe that all teeth were revascularized as determined by Doppler imaging after 2 years with no sign of endodontic/periodontal radiolucency. Doppler ultrasonic assessment of the periapical region of teeth never validate whether the dental pulp is revascularized as well as the vitality of the pulp tissue. Computed tomography images of teeth receiving pulp allogenic transplantation show that the dental pulps of all teeth suffer from obliteration, suggesting that the pulp tissue is not revascularized.
- All patients reported slight pain at the periapical region of recipient teeth up to 2-3 months after the procedure and a slight increase of periapical radiolucency occurred in two patients during 6 months to 1 year. The authors should discuss what happened at the periapical lesions of these teeth.
Author Response
First of all, the authors of this study would like to thank the Editor and referees of the Bioengineering MDPI for their constructive comments. The changes in the text are highlighted in yellow. Below you can find the answers to each comment of referees.
Reviewer 1
1.The authors describe that all teeth were revascularized as determined by Doppler imaging after 2 years with no sign of endodontic/periodontal radiolucency. Doppler ultrasonic assessment of the periapical region of teeth never validate whether the dental pulp is revascularized as well as the vitality of the pulp tissue. Computed tomography images of teeth receiving pulp allogenic transplantation show that the dental pulps of all teeth suffer from obliteration, suggesting that the pulp tissue is not revascularized.
Answer: Ultrasonic Doppler measures blood flow to a specific region of the body by the interaction of laser radiation. This technique allows an investigation of blood vessels in the periapical region, in the study of blood perfusion. In this regard, we surveyed the blood flow in the periapical region of each teeth that received allogenic transplantation. Indeed, the sum of the three outcomes (blood flow, electric pulp vitality test and tomographic images) suggests that the pulp tissue was revascularized. This was informed in the Discussion section. Therefore, the ultrasonic assessment reinforces further outcomes, and it has been validated in previous investigations (Cho YW, Park SH. Use of ultrasound Doppler to determine tooth vitality in a discolored tooth after traumatic injury: its prospects and limitations. Restor Dent Endod. 2014 Feb;39(1):68-73. doi: 10.5395/rde.2014.39.1.68.; Maity I, Meena N, Kumari RA. Single visit nonsurgical endodontic therapy for periapical cysts: A clinical study. Contemp Clin Dent. 2014 Apr;5(2):195-202. doi: 10.4103/0976-237X.132321.; Lima TFR, Dos Santos SL, da Silva Fidalgo TK, Silva EJNL. Vitality Tests for Pulp Diagnosis of Traumatized Teeth: A Systematic Review. J Endod. 2019 May;45(5):490-499. doi: 10.1016/j.joen.2019.01.014.).
2.All patients reported slight pain at the periapical region of recipient teeth up to 2-3 months after the procedure and a slight increase of periapical radiolucency occurred in two patients during 6 months to 1 year. The authors should discuss what happened at the periapical lesions of these teeth.
Answer: Almost all root canal instrumentation yield in slight pain due to the contact of irrigation solutions with bone and periodontal ligament. Furthermore, the explanation of initial periapical radiolucency was already included in Discussion section as follows: “Conversely, in the current clinical cases, all patients presented periapical radiolucency before or few months after the allogenic transplantation. Such a feature may be a consequence of lower blood irrigation along with small inflammatory mediators due to the presence of allogenic cells, which may interfere with the success of the treatment. Nevertheless, successful outcomes were attained in the current study, along with a clear reduction of the periapical lesion in the treated teeth”.

Reviewer 2 Report
This is an interesting paper showed in very limited cases; transplantation of pulp tissue from offspring may encourage periapical healing of endodontically affected single rooted teeth in parents. There are several concerns with the methodology and conclusions of this study that are listed below:
- As it seems, the total hemostasis was achieved in recipient teeth, can authors explain possible mechanism of pulpal vascularization of transplanted pulpal tissue.
- It seems there is no control in this case study. There is evidence that temporary healing within the duration of authors study can be achieved by simple removing infected pulp, sterilizing the pulp chamber and sealing it with restorative materials. Without any controls, it is hard to contribute observed healing solely to the presence of the transplanted pulp.
- The results of the vitality tests with electric pulp testers may not be reliable. There is also no quantitative value given for the range of responses. Such tests should have been also performed on recipient teeth perhaps prior to treatment to role out any false positive readings.
- There is no evidence of the health status of the transplanted pulp tissue. A simple cell culture method or histochemical staining for cell apoptosis could verify the health status of transplanted tissue.
- I am not sure how ethical it is to expose patients to repeated radiations of CT at 6, 12 and 24 months, as well as to repeated periapical radiographies at 3, 9, 15 and 18 months for only the purpose of this research?
Author Response
First of all, the authors of this study would like to thank the Editor and referees of the Bioengineering MDPI for their constructive comments. The changes in the text are highlighted in yellow. Below you can find the answers to each comment of referees.
Reviewer 2
1.As it seems, the total hemostasis was achieved in recipient teeth, can authors explain possible mechanism of pulpal vascularization of transplanted pulpal tissue.
Answer: The main types of stem cells in dental pulp are identified as mesenchymal stem cells, which has the proliferative and regenerative capacity towards different tissues. Along with these cells, vascular and neuronal growth factors play a key role in such homeostasis and regeneration process. This factors are already in stem cell and dentinal tubules. Such explanation was already included in Discussion section.
2.It seems there is no control in this case study. There is evidence that temporary healing within the duration of authors study can be achieved by simple removing infected pulp, sterilizing the pulp chamber and sealing it with restorative materials. Without any controls, it is hard to contribute observed healing solely to the presence of the transplanted pulp.
Answer: The referee is correct about such evidence. However, this treatment would provide healing of periapical lesion without pulp/tooth vitality recovery. The same is expected with traditional treatment using gutta percha as filling material. In this case series, we presented a completely new clinical technique able to attain revitalization along with healing. The proper controls will be included in the clinical trial.
3.The results of the vitality tests with electric pulp testers may not be reliable. There is also no quantitative value given for the range of responses. Such tests should have been also performed on recipient teeth perhaps prior to treatment to role out any false positive readings.
Answer: We agree with the referee. Nevertheless, all patient looked for care with spontaneous pain in receptor teeth, thus, the electric test could not be performed at that point. For control purposes, intact teeth from same patients were surveyed as part of the analysis. This information was included in Materials and Methods section.
4.There is no evidence of the health status of the transplanted pulp tissue. A simple cell culture method or histochemical staining for cell apoptosis could verify the health status of transplanted tissue.
Answer: Histochemical staining and cell culture of these transplanted pulps would be possible only after extraction of the teeth. However, the teeth are still in function and health, there is no clinical reason to support extraction for the sake of in vitro tests. Indeed, the ultrasonic doopler assessment showed revascularization, which was positive and could be correlated to health pulp tissues.
5.I am not sure how ethical it is to expose patients to repeated radiations of CT at 6, 12 and 24 months, as well as to repeated periapical radiographies at 3, 9, 15 and 18 months for only the purpose of this research?
Answer: Indeed, patients could be exposed to too much radiation. However, a likely rejection of transplants and acute reaction in bone could be harmful to the patients. Therefore, the Ethics Committee has approved the protocol.
We hope these changes turn the paper acceptable for publication.
Best regards,
Authors

Round 2
Reviewer 1 Report
Feitosa V et al.: The allogenic dental pulp transplantation from son/daughter to 2 mother/father: a follow-up of three clinical cases
The authors failed to improve the manuscript according to this reviewer’s first-round criticisms. Of course, the reviewer understands that ultrasonic Doppler measures blood flow to a specific region of the body by the interaction of laser radiation. The criticism raised by the reviewer is how the authors validated that the periapical region of teeth was properly measured. Similarly, the response by the authors in terms of the second criticism is insufficient. Consequently, this reviewer regrets to inform the authors that it was found unsuitable for publication in Bioengineering.
Major concerns:
- The authors describe that all teeth were revascularized as determined by Doppler imaging after 2 years with no sign of endodontic/periodontal radiolucency. Doppler ultrasonic assessment of the periapical region of teeth never validate whether the dental pulp is revascularized as well as the vitality of the pulp tissue. Computed tomography images of teeth receiving pulp allogenic transplantation show that the dental pulps of all teeth suffer from obliteration, suggesting that the pulp tissue is not revascularized.
- All patients reported slight pain at the periapical region of recipient teeth up to 2-3 months after the procedure and a slight increase of periapical radiolucency occurred in two patients during 6 months to 1 year. The authors should discuss what happened at the periapical lesions of these teeth.
Reviewer 2 Report
Thanks for the answers to the questions.
The question 4 "There is no evidence of the health status of the transplanted pulp tissue. A simple cell culture method or histochemical staining for cell apoptosis could verify the health status of transplanted tissue" actually refers to the health status of the transplanted tissue not the recipient tooth. Histochemical staining and /or cell culture needed to verify the health status of the transplanted tissue. This is should be included as a part of future clinical trial.
A paragraph regarding repeatability and reliability of electric pulp testing in context of this experiment should be included in discussion.